# Simultaneous Induction of Glycolysis and Oxidative Phosphorylation during Activation of Hepatic Stellate Cells Reveals Novel Mitochondrial Targets to Treat Liver Fibrosis

**DOI:** 10.3390/cells9112456

**Published:** 2020-11-11

**Authors:** Natalia Smith-Cortinez, Karen van Eunen, Janette Heegsma, Sandra Alejandra Serna-Salas, Svenja Sydor, Lars P. Bechmann, Han Moshage, Barbara M. Bakker, Klaas Nico Faber

**Affiliations:** 1Department of Gastroenterology and Hepatology, University Medical Center Groningen, University of Groningen, 9712 CP Groningen, The Netherlands; natalia.fsc@gmail.com (N.S.-C.); j.heegsma@umcg.nl (J.H.); s.a.serna.salas@umcg.nl (S.A.S.-S.); a.j.moshage@umcg.nl (H.M.); 2Department of Pediatrics, University Medical Center Groningen, University of Groningen, 9712 CP Groningen, The Netherlands; k.van.eunen@umcg.nl (K.v.E.); b.m.bakker01@umcg.nl (B.M.B.); 3Department of Internal Medicine, University Hospital Knappschaftskrankenhaus, Ruhr-University, 44892 Bochum, Germany; svenja.sydor@ruhr-uni-bochum.de (S.S.); lars.bechmann@rub.de (L.P.B.)

**Keywords:** cell metabolism, liver fibrosis, mitochondria

## Abstract

Upon liver injury, hepatic stellate cells (HSCs) transdifferentiate to migratory, proliferative and extracellular matrix-producing myofibroblasts (e.g., activated HSCs; aHSCs) causing liver fibrosis. HSC activation is associated with increased glycolysis and glutaminolysis. Here, we compared the contribution of glycolysis, glutaminolysis and mitochondrial oxidative phosphorylation (OXPHOS) in rat and human HSC activation. Basal levels of glycolysis (extracellular acidification rate ~3-fold higher) and particularly mitochondrial respiration (oxygen consumption rate ~5-fold higher) were significantly increased in rat aHSCs, when compared to quiescent rat HSC. This was accompanied by extensive mitochondrial fusion in rat and human aHSCs, which occurred without increasing mitochondrial DNA content and electron transport chain (ETC) components. Inhibition of glycolysis (by 2-deoxy-D-glucose) and glutaminolysis (by CB-839) did not inhibit rat aHSC proliferation, but did reduce *Acta2* (encoding α-SMA) expression slightly. In contrast, inhibiting mitochondrial OXPHOS (by rotenone) significantly suppressed rat aHSC proliferation, as well as *Col1a1* and *Acta2* expression. Other than that observed for rat aHSCs, human aHSC proliferation and expression of fibrosis markers were significantly suppressed by inhibiting either glycolysis, glutaminolysis or mitochondrial OXPHOS (by metformin). Activation of HSCs is marked by simultaneous induction of glycolysis and mitochondrial metabolism, extending the possibilities to suppress hepatic fibrogenesis by interfering with HSC metabolism.

## 1. Introduction

Liver fibrosis is characterized by the intrahepatic accumulation of connective tissue as a consequence of chronic liver injury. This process occurs in most types of chronic liver diseases, including immune-, viral-, alcoholic- and metabolic-mediated forms [1]. Liver fibrosis may progress to cirrhosis, where the liver architecture is disrupted due to the accumulation of extracellular matrix (ECM) and may ultimately lead to liver failure [1]. Moreover, patients with cirrhosis are at increased risk to develop hepatocellular carcinoma. Liver transplantation is then the only available therapy left [2]. Fortunately, accumulating evidence reveals that liver fibrosis in early stages is largely reversible [2] and major efforts are being made to develop new drug-based therapies to reverse liver fibrosis and/or prevent progression to cirrhosis.

The main cell type involved in liver fibrosis is the hepatic stellate cell (HSC). HSCs are responsible for producing most of the ECM proteins, like collagens and fibronectins that accumulate in the chronically injured liver. HSCs in the healthy liver are considered “quiescent” (qHSCs), reside in the space of Disse, store vitamin A in large cytoplasmic lipid droplets and express high levels of the lipogenic transcription factor peroxisome proliferator-activated receptor gamma (PPAR-γ) [3]. The qHSCs contain the largest reserves of vitamin A in the human body and are considered key regulators of systemic distribution to vitamin A-requiring peripheral tissues [4]. Upon liver injury, however, qHSCs lose their vitamin A and lipid stores, turn down PPAR-γ expression and transdifferentiate into myofibroblast-like cells (e.g., activated HSCs (aHSCs)) that are highly proliferative, contractile, migratory and produce excessive amounts of extracellular matrix proteins, including collagen I and III [5]. The aHSCs express alpha-smooth muscle actin (α-SMA) and produce pro-inflammatory and pro-fibrogenic cytokines, like TGF-β [3,6]. Hepatic stellate cell activation is central to the development of fibrosis and thus an important process to target in the quest to treat liver fibrosis. 

Furthermore, inhibiting these metabolic pathways has been shown to prevent the undergoing activation in vitro [7,8]. It has been recently demonstrated that plastic-cultured aHSCs have enhanced oxidative phosphorylation and glycolysis compared to less activated matrigel-cultured aHSCs [9]. However, functional analyses to quantify metabolic processes that differentiate freshly isolated qHSCs versus culture-activated HSCs have not been reported yet. Here, we hypothesize that HSC transdifferentiation is not only accompanied by a shift to glycolysis and glutaminolysis, but also an induction of mitochondrial oxidative phosphorylation (OXPHOS) to meet the high energy demand of aHSCs. Thus, the objectives of this study were to investigate the contribution of mitochondrial tic metabolism in the activation process of HSCs and identify new mitochondrial targets for the treatment of liver fibrosis.

## 2. Materials and Methods

### 2.1. Reagents

Fully activated rat and human HSCs (at least 7 days in culture) were treated for 3 days with 2.5 mmol/L 2-Deoxy-D-glucose (2DG, D8375, Sigma, Saint Luis, MO, USA) or 5 µmol/L GLS1 inhibitor (CB-839, 5.33717, EMD Millipore, Burlington, NJ, USA) to inhibit glycolysis or glutaminolysis, respectively. OXPHOS was inhibited by either 5 µmol/L rotenone (R8875, Sigma) or 2 mmol/L metformin (317240, EMD Millipore).

### 2.2. Rat and Human HSC Isolation, Cell Culture and Treatments

Primary rat hepatic stellate cells were isolated by a two-step perfusion of the liver with pronase (Merck, Amsterdam, The Netherlands) and collagenase-P (Roche, Almere, The Netherlands) and further purified by Nycodenz (Axis-ShieldPOC, Oslo, Norway) gradient centrifugation, as described before [10]. HSCs were cultured in Iscove’s modified Dulbecco’s medium (IMDM) with Glutamax (Invitrogen, Breda, The Netherlands) supplemented with 20% heat-inactivated fetal calf serum (FCS, Invitrogen), 1 mmol/L sodium pyruvate (Invitrogen), 1x MEM non-essential amino acids (Invitrogen), 50 µg/mL gentamicin (Invitrogen) and penicillin–streptomycin–fungizone (PSF) in a humidified incubator at 37 °C with 5% CO_2_. Freshly isolated quiescent rat HSCs (r-qHSCs) were allowed to attach to culture plates for 4 or 24 h and harvested (r-qHSCs), or cultured for 7 days to become fully activated rHSCs (r-aHSCs). After 7 days, cells were trypsinized and seeded for experiments. 

Primary human HSCs were isolated from macroscopically normal liver specimens obtained from fresh tumor resections by an “all-in-one” liver cell purification procedure, as described previously [11]. Freshly isolated quiescent human HSCs were cultured for two weeks as part of the purification protocol and to become fully activated HSCs (h-aHSCs). h-aHSCs were passaged by trypsinization and passages two to five were used for experiments. 

LX-2 cells were obtained from Merk Millipore (SCC064) and used in passages 22–30. Dulbecco’s modified Eagle’s medium (DMEM) with high glucose supplemented with 10% fecal calf serum and 1% antibiotics was used for culturing them. Cells were passed by trypsinization and medium was refreshed every 3 days or when necessary. Cells were grown in 5% CO_2_ and 37 °C in ambient air.

### 2.3. Real-Time Monitoring of Cell Proliferation

Proliferation was assessed using the xCELLigence Real Time Cell Analyzing (RCTA) system (RTCA DP; ACEA Biosciences, Inc., San Diego, CA, USA). Primary human or rat aHSCs were plated in E-plates to record cell proliferation, according to the manufacturer’s instructions [12]. Treatment was started 24 h after attachment and finished after an additional 72 h of treatment. Results were recorded and analyzed by RTCA software.

### 2.4. Real-Time Imaging Monitoring of Cell Dynamics

Cell migration, morphology and proliferation were assessed by the Live-Cell Analysis System Incucyte (Sartorius, Biopharma, Göttingen, Germany). Primary human or rat activated HSCs were seeded and after 24 h of attachment cells were treated and taken to the IncuCyte ZOOM^®^ platform, which was housed inside a cell incubator at 37 °C/7.5% CO_2_, for 72 h. Nine images per well from three technical replicates were taken every 2 h using a 10× objective lens and then analyzed using IncuCyte™ Basic Software.

### 2.5. RNA Isolation, cDNA Synthesis and Real-Time Quantitative PCR

A quantitative real-time reverse transcription polymerase chain reaction (qRT-PCR) was performed as previously described [13]. Briefly, RNA was isolated from cell cultures using TRIzol^®^ reagent according to the supplier’s instructions (Thermo Fisher Scientific, Breda, The Netherlands). RNA quality and quantity were determined using a NanoDrop 2000c UV–vis spectrophotometer (Thermo Fisher Scientific). The cDNA was synthesized from 2.5 µg RNA using random nanomers and M-MLV reverse transcriptase (Invitrogen, Carlsbad CA, USA). Taqman primers and probes were designed using Primer Express 3.0.1 and are shown in Appendix A. All target genes were amplified using the Q-PCR core kit master mix (Eurogentec, Liège, Belgium) on a 7900HT Fast Q-PCR system (Thermo Fisher Scientific). SDSV2.4.1 (Thermo Fisher Scientific) software was used to analyze the data. Expression of genes is presented in 2-delta CT and normalized to 18S.

### 2.6. Protein Isolation, Quantification and Western Blot Analysis

Protein samples were prepared for Western blot analysis as described previously [14]. Protein concentrations were quantified using the Bio-Rad protein assay (Bio-Rad, Hercules, CA, USA) with bovine serum albumin (BSA) as a standard. Thirty micrograms of protein were separated on Mini-PROTEAN^®^ TGX™ precast 4–15% gradient gels (Bio-Rad) and transferred to nitrocellulose membranes using the Trans-Blot Turbo transfer system, (Bio-Rad). Primary antibody details and dilutions are listed in Appendix A and appropriate horseradish peroxidase (HRP)-conjugated secondary antibodies (1:2000; P0448, DAKO) were used for detection. Proteins were detected using the Pierce ECL Western blotting kit (Thermo Fisher Scientific). Images were captured using the ChemiDoc XRS system and Image Lab version 3.0 (Bio-Rad). 

### 2.7. Quantification of Glycolysis and OXPHOS

Extracellular acidification rates (ECARs) and oxygen consumption rates (OCRs) were measured using glycolysis stress kits and mito stress kits, respectively, in the extracellular flux analyzer Seahorse (Agilent Technologies, Santa Clara, CA, USA) following the manufacturer’s instructions and as described previously (14). Briefly, respiratory chain inhibitors (Oligo: oligomycin, a complex V inhibitor, FCCP: carbonyl cyanide-p-trifluoromethox- yphenyl-hydrazon, a protonophore and lastly A+R: antimycin A and rotenone, inhibitors of complex III and I) and glycolysis stimulators/inhibitors (glucose, to stimulate glycolysis, oligomycin, to inhibit mitochondrial metabolism and help reach glycolysis to its maximum, and 2-Deoxyglucose, glycolysis inhibitor) were added sequentially as described previously [15] and following manufacturer’s instructions. OCR and ECAR were measured at 37 °C.

### 2.8. Enzyme Activity and Capacity (Vmax)

Enzymatic activity assays were performed in cellular extracts and all parameters were adapted from yeast to mammalian cells, as previously described [16]. Briefly, all enzyme activities were measured in freshly prepared extracts at 37 °C in a Synergy H4 plate reader (BioTek, Winooski, VT, USA) as described previously [17]. For all assays, the reaction mixtures without start reagent were pre-warmed at 37 °C. The conditions used for each enzymatic reaction are described here: Hexokinase (HK; EC2.7.1.1)—1.2 mmol/L NADP+, 10 mmol/L glucose, 1.8 U/mL glucose-6-phosphate dehydrogenase (EC1.1.1.49), and 10 mmol/L ATP as start reagent. Phosphoglucose isomerase (GPI; EC5.3.1.9)—0.4 mmol/L NADP+, 1.8 U/mL glucose-6-phosphate dehydrogenase (EC1.1.1.49), and 2 mmol/L fructose 6-phosphate as start reagent. Phosphofructokinase (PFK; EC2.7.1.11)—0.15 mmol/L NADH, 1 mmol/L ATP, 0.5 U/mL aldolase (EC4.1.2.13), 0.6 U/mL glycerol-3P-dehydrogenase (EC1.1.1.8), 1.8 U/mL triosephosphate isomerase (EC5.3.1.1), and 10 mmol/L fructose 6-phosphate as start reagent. Aldolase (ALD; EC4.1.2.13)—0.15 mmol/L NADH, 0.6 U/mL glycerol-3P-dehydrogenase (EC1.1.1.8), 1.8 U/mL triosephosphate isomerase (EC5.3.1.1), and 2 mmol/L fructose 1,6-bisphosphate as start reagent. Glyceraldehyde-3-phosphate dehydrogenase (GAPDH; EC1.2.1.12)—0.15 mmol/L NADH, 1 mmol/L ATP, 24 U/mL 3-phosphoglycerate kinase (EC2.7.2.3), and 5 mmol/L 3-phosphoglyceric acid as start reagent. 3-Phosphoglycerate kinase (PGK; EC2.7.2.3)—0.15 mmol/L NADH, 1 mmol/L ATP, 8 U/mL glyceraldehyde-3-phosphate dehydrogenase (EC1.2.1.12), and 5 mmol/L 3-phosphoglyceric acid as start reagent. Phosphoglycerate mutase (PGAM; EC5.4.2.1)—0.15 mmol/L NADH, 1 mmol/L ADP, 2.5 mmol/L 2,3-diphospho-glyceric acid, 5 U/mL enolase (EC4.2.1.11), 50 U/mL pyruvate kinase (EC2.7.1.40), 60 U/mL L-lactate dehydrogenase (EC1.1.1.27), and 5 mmol/L 3-phosphoglyceric acid as start reagent. Enolase (ENO; EC4.2.1.11)—0.15 mmol/L NADH, 1 mmol/L ADP, 50 U/mL pyruvate kinase (EC2.7.1.40), 15 U/mL L-lactate dehydrogenase (EC1.1.1.27), and 1 mmol/L 2-phosphoglyceric acid as start reagent. Pyruvate kinase (PK)—0.15 mmol/L NADH, 1 mmol/L ADP, 1 mmol/L fructose 1,6-bisphosphate, 60 U/mL L-lactate dehydrogenase (EC1.1.1.27) and 2 mmol/L phosphoenolpyruvate as start reagent. Lactate dehydrogenase (LDH)—0.15 mmol/L NADH, and 1 mmol/L pyruvate as start reagent. Glucose-6-phosphate dehydrogenase (G6PDH)—0.4 mmol/L NADP+, and 5 mmol/L glucose 6-phosphate as start reagent. 

### 2.9. ATP Assay

Primary rat HSCs were seeded in black 96-well plates and after 24 h of attachment, cells were treated. After 72 h, cells were lysed according to the manufacturer’s instructions (Cell-Triter GLO, Promega, Madison, WI, USA) and luminescence was recorded in a Synergy H4 (BioTek) plate reader at RT.

### 2.10. Immunofluorescence Microscopy

Primary rat HSCs were seeded directly after isolation in 12-well plates containing 18 mm glass coverslips. Cells on coverslips were washed, fixed (4% PFA, 10 min) and permeabilized (0.1% Triton X-100, 10 min) prior non-specific blocking (2% BSA, 30 min). After blocking, cell on coverslips were incubated for 1 h at RT with the primary antibodies goat polyclonal collagen-I (Southern Biotech, 1310-01, 1/200) and mouse monoclonal αSMA (Sigma Aldrich (Munich, Germany), A5228, 1/100), washed three times with blocking solution and incubated with secondary antibodies goat anti-rabbit Alexa Fluor 488 or rabbit anti-mouse Alexa Fluor 488 (Thermo Fisher Scientific, 30 min, at RT). After secondary antibody incubation, coverslips were washed three times and mounted with Vectashield Antifade Mounting Medium with DAPI (Vector Laboratories, Gdynia, Poland). Coverslips were air-dried, sealed using nail polish, stored at 4 °C and covered from light until further use. Images were obtained in a Zeiss 410 inverted laser scan microscope (Leica Microsystems, Wetzlar, Germany) with 16× or 40× magnification objectives using immersion oil and processed using ImageJ software (public domain, developed at the National Institutes of Health).

### 2.11. Statistical Analysis

All data are presented as mean ± standard deviation. Significance of differences between groups was tested by one-way ANOVA and *t*-tests. Calculations were made using the software GraphPad Prism 5. Results were considered statistically different with *p* value < 0.05. 

## 3. Results

### 3.1. Glycolytic and Mitochondrial Metabolism Are Increased in Activated HSCs

The extracellular acidification rate (ECAR, measure of glycolysis) and oxygen consumption rate (OCR, measure of mitochondrial OXPHOS) were quantified in quiescent primary rat r-qHSCs (day 1) and fully activated r-aHSCs (day 7) using a Seahorse extracellular flux analyzer (Figure 1). Rat qHSCs hardly show any non-glycolytic acidification, while this is readily detectable for r-aHSCs (Figure 1A, left panel level after 2DG addition; right panel last bar graph). Basal glycolytic activity was significantly higher (3-fold) in r-aHSCs, when compared to r-qHSCs (Figure 1A, 7.6 ± 2.7 mpH/min/105 cells in r-qHSCs versus 23.4 ± 3.5 mpH/min/105 cells in r-aHSCs; *p* < 0.0001). Glycolytic capacity was significantly higher in r-aHSCs compared to r-qHSCs (Figure 1A, 6.2 ± 1.4 mpH/min/105 cells in r-qHSCs versus 40.5 ± 2.1 mpH/min/105 cells in r-aHSCs; *p* < 0.01). As a consequence, r-aHSCs showed a significantly higher reserve glycolytic capacity compared to r-qHSCs (Figure 1A, 8.5 ± 5.2 versus 30.84 ± 4.9 mpH/min/105 cells, respectively; *p* < 0.01). Basal mitochondrial respiration was even more pronouncedly enhanced (4.8-fold) in r-aHSCs compared to r-qHSCs (Figure 1C, 305.9 ± 69.0 versus 63.1 ± 4.3 pmol O2/min/105 cells, respectively; *p* < 0.0001), which was almost completely attributed to ATP-linked respiration (Figure 1C, 249.2 ± 48.0 versus 27.7 ± 2.5 pmol O2/min/105 cells, respectively; *p* < 0.0001). Moreover, r-aHSCs also harbored significantly more spare respiratory capacity, which culminated in a significantly higher maximum respiration capacity of r-aHSCs compared to r-qHSCs (Figure 1C, 436.4 ± 92 versus 121.5 ± 15 pmol O2/min/105 cells, respectively; *p* < 0.0001). Non-mitochondrial respiration and proton leak, a measure of mitochondrial coupling efficiency [18], were similar for r-qHSCs and r-aHSCs (Figure 1C). 

Quantification of the activity of 11 glycolytic enzymes revealed only a significant increase in phosphoglycerate kinase (PGK, +50%) and glucose-6-phosphate dehydrogenase (G6PDH + 300%) activity after rHSC activation (Figure 1B). All other enzymes showed similar activities in r-qHSCs and r-aHSCs, though non-significant increases were noticed for most of them in r-aHSCs. Of note, the normalization in ECAR analysis was performed per cell number and in glycolytic activity per mg protein. Taken together, these results suggest a functional upregulation of mitochondrial and glycolytic metabolism to a highly energetic phenotype (Figure 1D) to support r-aHSC growth, proliferation, motility and ECM production.

### 3.2. Mitochondrial Fusion Is Increased during Activation of HSCs

Upregulation of mitochondrial metabolism has been linked to mitochondrial fusion in various cell types [19,20]. To assess mitochondrial fusion, immunofluorescence of freshly isolated r-qHSCs and r-aHSCs (3 and 7 days after isolation) was performed. Antibodies against manganese superoxide dismutase (MnSOD) were used to visualize mitochondria and rHSC activation was assessed by α-SMA immunostaining. Figure 2A shows that MnSOD-containing mitochondria are mostly observed as individual dots in quiescent HSCs, distributed along the periphery of the cells (Figure 2A, left panel), while α-SMA-specific staining was not detected, as expected. Very strong staining of the α-SMA network was detected in r-aHSCs, cells that had also strongly increased in size (Figure 2A, middle panel (day 3) and right panel (day 7)). This was associated with a highly fused MnSOD-stained mitochondrial network, distributed in close proximity to the nuclei and extended towards the cell projections (Figure 2A, middle and right panels) An interconnected mitochondrial phenotype thus correlates with a highly energetic state of aHSCs (Figure 1D). Immunostaining of human aHSCs with MnSOD and mitofusin 1 (MFN1, as a marker of mitochondrial fusion) revealed that mitochondria are highly fused (as seen by the high co-localization of MnSOD and MFN1) and are distributed in the perinuclear and extended to the cell extensions (Figure 2B) similar to that seen in rat aHSCs.

Interestingly, LX-2 cells, used as a model of immortalized human HSCs, stained with MitoTracker red, showed that mitochondria are also highly fused (Figure 2C). Mitochondrial fusion protein MFN1 and MnSOD co-staining in r-qHSCs revealed that these two proteins do not co-localize, confirming that MFN1 is not playing a role in mitochondrial fusion in rat qHSCs (Figure 2D).

### 3.3. Increase in Mitochondrial Fusion during Activation Is Independent of Mitochondrial Biogenesis or Mitochondrial Copy Number

We next determined whether the increase in mitochondrial metabolism and fusion was due to increased mitochondrial biogenesis and/or mitochondrial mass. Protein and mRNA levels of the peroxisome proliferator-activated receptor gamma coactivator 1 alpha (PGC1-α), a key controller of mitochondrial biogenesis, were readily detectable in r-qHSCs (day 0 and 1 after isolation; Figure 3A,E). However, PGC1-α protein levels rapidly dropped upon activation of rHSCs (day 3 and day 7), concomitant with a reduction in the corresponding *Ppargc1a* mRNA levels (Figure 3A,E). Interestingly, protein levels of the voltage-dependent anionic channel (VDAC), located in the outer membrane of mitochondria, were also strongly downregulated upon rHSC activation (Figure 3A). Activation of rHSCs was confirmed by a strong induction of α-SMA protein/*Acta2* mRNA levels (Figure 3A,E), in conjunction with increased *Col1a1* levels (Figure 3E) and reduced Nr1c3 (encoding PPAR-γ) levels (Figure 3E). Analysis MFN1 during r-HSC activation showed that it was readily detected in r-qHSCs (as also observed by immunofluorescence, Figure 2D), and after an early drop in expression during HSC activation on day 1 and 3, levels rose again in fully activated r-HSCs (Figure 3A). Analysis of mRNA levels of mitofusin 1 and 2 (*Mfn1* and *Mfn2*) revealed slight upregulation on day 1 after isolation and a slight decrease on day 7 after isolation (Figure 3E). Analysis of the dynamin-related protein 1 (DRP1) revealed that only 7 days after isolation, this marker was upregulated at protein levels (Figure 3A). Interestingly, the corresponding mRNA levels of *Dnm1l* did not significantly change during rHSC activation (Figure 3E). The combined mRNA and protein analysis reveals that levels of factors involved in mitochondrial dynamics appear to be regulated partly in a post-transcriptional manner during the HSC activation process. Western blot analysis of OXPHOS proteins revealed a clear decrease in complex III only in day 7 r-aHSCs (Figure 3B), while levels of other components were rather comparable at the different time points of rHSC activation. Moreover, copy numbers of mitochondrial DNA (as a proxy for mitochondrial mass) did not change during the rHSC activation process (Figure 3C). Interestingly, MFN1, DRP1 and phosphorylated DRP1 (p-DRP1) expression was also readily detected in h-aHSCs as in a-rHSCs, while those proteins were evidently lower in r-qHSCs (Figure 3E). Taken together, these results show that the increase in mitochondrial metabolism (Figure 1) and mitochondrial fusion (Figure 2) are independent of mitochondrial biogenesis and/or increase in mass.

### 3.4. Mitochondrial and Glycolytic Metabolism Are Independently Necessary to Maintain the Activated State of HSCs

We next made a side-by-side comparison of glycolysis inhibition, glutaminolysis inhibition or complex I of the mitochondrial electron transport chain (ETC) inhibition in r-aHSCs and h-aHSCs. We inhibited glycolysis (with 2DG), glutaminolysis (with CB-839) or complex I (with rotenone in rat and metformin in human aHSCs) in r-aHSCs (Figure 4) and h-aHSCs (Figure 5). It is important to note that none of the treatments led to significant levels of cell death after 72 h (Appendix A). Interestingly, inhibition of glycolysis caused a significant reduction in *ACTA2* (encoding α-SMA) and *COL1A1* (encoding collagen I) mRNA levels only in human aHSCs (Figure 5A) with no effect in r-aHSCs (Figure 4A). In line with this, inhibiting glycolysis significantly suppressed h-aHSCs proliferation (Figure 5B; 66.4% ± 8.7 reduction, *p* < 0.0001), while this was not observed for r-aHSCs (Figure 4C). Interestingly, 2DG treatment caused a significant reduction in cellular ATP levels in r-aHSCs (Figure 4B). Similar as observed for glycolysis inhibition, targeting glutaminolysis also had differential effects in r-aHSCs and h-aHSCs; while *ACTA2/Acta2* levels were suppressed by CB-839 treatment in both r-aHSCs and h-aHSCs, *COL1A1* levels were only reduced in h-aHSCs (Figure 4A and Figure 5A). Moreover, proliferation and cellular ATP levels were not affected by CB-839 in r-aHSCs (Figure 4B,C), while it significantly reduced h-aHSC proliferation (Figure 5B; 46.9 ± 67.5 reduction, *p* < 0.001). Importantly, inhibition of mitochondrial OXPHOS suppressed mRNA levels of *ACTA2/Acta2* and *COL1A1/Col1a1* in both (rotenone-treated) r-aHSCs and (metformin-treated) h-aHSCs (Figure 4A and Figure 5A), and was accompanied by impaired cell proliferation (Figure 4C and Figure 5B) and a reduction in cellular ATP concentrations (Figure 4B) Taken together, these results show that besides glycolysis, mitochondrial metabolism also supports HSC activation and when inhibited cannot be fully compensated by other energy-generating pathways. 

## 4. Discussion

In this study, we show that HSC transdifferentiation is characterized by simultaneous induction of glycolysis and mitochondrial oxidative phosphorylation (OXPHOS) and is accompanied by extensive mitochondrial fusion to meet the high energy demands associated with induced cell proliferation, migration, contraction, migration and ECM production. The HSC transdifferentiation-associated induction of OXPHOS (~5-fold) was stronger than glycolysis (~3-fold). HSC activation was suppressed by targeting glycolysis or mitochondrial metabolism, either glutaminolysis or OXPHOS, separately, but most pronounced by inhibiting OXPHOS. Human activated HSCs appear to be more sensitive to metabolic inhibitors compared to activated rat HSCs. Thus, metabolic shifts associated with HSC transdifferentiation reveal novel and potent targets for the treatment of liver fibrosis.

The first question this study sought to determine was the actual contribution of glycolysis and oxidative phosphorylation towards HSC activation by comparing freshly isolated versus culture-activated primary HSCs. Several reports have shown that HSCs depend on glycolysis and glutaminolysis to transdifferentiate towards active HSCs [7,8]. These reports analyzed protein and mRNA expression of key enzymes in glycolysis and glutaminolysis during rat HSC activation [7,8]. However, actual glycolytic rat and/or mitochondrial respiration were not quantified. Interestingly, others have shown that plastic-cultured r-aHSCs and LX-2 cells have higher mitochondrial and glycolytic metabolism compared to matrigel-cultured r-aHSCs and LX-2 cells [9]. Here, we show that the glycolytic rate is increased approximately 3-fold in culture-activated rHSCs versus freshly isolated r-qHSCs. Surprisingly though, the mitochondrial oxygen consumption rate increased even more, by about 5-fold. This is the first time that the simultaneous activation of glycolysis and mitochondrial metabolism in culture-activated versus freshly isolated HSCs is reported. Interestingly, the induction of mitochondrial OXPHOS was accompanied by extensive mitochondrial fusion, without changes in mitochondrial mass. Mitochondria play a key role in regulating many different processes like calcium homeostasis, generation and control of reactive oxygen species, regulation of programmed cell death and ATP production [21]. The mitochondrial morphology is highly linked to its functions. Mitochondrial fragmentation occurs in response to nutrient excess and cellular dysfunction and mitochondrial fusion is associated with increased cell bioenergetic demands [19]. This suggests that the increase in mitochondrial fusion is a direct response to the increasing energetic demand during HSC activation. Surprisingly, the observed increase in mitochondrial respiration (5-fold) and mitochondrial fusion was not accompanied by elevated levels of mitochondrial ETC complex proteins. This might be explained by the fact that the mitochondrial ETC proteins form super complexes in response to high energy demands [22] to create a more efficient ETC machinery without upregulating total protein levels. It needs to be noted though that the normalization in the two methods, e.g., mitochondrial respiration per cell number and ETC proteins per total cellular protein, is different. HSCs grow enormously in size, and thus protein content, during activation in vitro, and the absolute amount of mitochondrial ETC protein increase along with it. Still, mitochondrial DNA copy number did not increase during HSC transdifferentiation. Thus, it will be interesting to analyze whether ETC super complexes indeed are formed in activating HSCs to accommodate the high energy demands for cell proliferation, migration, contraction and ECM production. Interestingly, mitochondrial fusion was also observed in human primary culture-activated HSCs and in LX-2 cells, suggesting it is a well-conserved process. Unfortunately, we were not able to compare mitochondrial morphology and activity in isolated human qHSCs versus aHSCs, as the isolation and purification protocol of human HSCs from resected liver tissue includes a culturing phase where plastic-adherent HSCs are enriched over other contaminating hepatic cell types (such as hepatocytes and endothelial cells). The human HSCs rapidly activate in this culturing step, precluding a direct comparison of qHSCs versus aHSCs.

We observed a strong downregulation of PGC1-α and VDAC during the activation of HSCs, suggesting a decrease in mitochondrial biogenesis and mass. PGC1-α is a co-activator of PPARγ, which is a transcription factor that is markedly downregulated during HSC activation. However, the role of PGC1-α in mitochondrial biogenesis is mostly driven by its role as a co-activator of nuclear respiratory factor 2 (NRF2) [23]. Given the fact that PPARγ is virtually absent in aHSCs, the reduced levels of PGC1-α may therefore remain sufficient to activate NRF2 and maintain mitochondrial copy number. The accompanying reduction in VDAC protein levels during HSC activation might be due to a differential stability and/or turnover of this protein compared to total mitochondrial turnover.

To unravel new metabolic targets to treat liver fibrosis, we inhibited glycolysis (by 2DG), as well as two mitochondrial metabolic pathways, i.e., glutaminolysis (by CB-839) and OXPHOS (by rotenone and metformin). Inhibiting glycolysis suppressed cell proliferation and expression of fibrogenic markers in fully activated human HSCs. Surprisingly, though, this was not observed for fully activated rat HSCs. The latter result seems to contradict an earlier report [7] that observed significant anti-proliferative and anti-fibrogenic effects of 2DG in rat HSCs [7]. In that study, 2DG treatments were started 3 days after rat primary HSC isolation followed by an exposure to 2DG for an additional 3 days. Three day-cultured rat HSCs are, however, not fully activated yet. In contrast, we started 2DG treatment 7 days after isolation, a stage where maximum activation is achieved and reversal, rather that prevention, of the cell proliferation and fibrogenic phenotype of HSCs can be established. This may suggest that inhibiting glycolysis during the activation indeed suppresses fibrogenesis, but once full HSC activation is reached, 2DG is not really effective in reversing the fibrogenic phenotype of rat HSCs. Interestingly, 2DG treatment in r-aHSCs decreased cellular ATP levels, however, this was not sufficient to suppress cell proliferation and activation markers in r-aHSCs. Still, 2DG did suppress cell proliferation and expression of fibrogenic markers in fully activated human HSCs which indicates that this may be species dependent. Unfortunately, inhibitors of glycolysis that have potent “therapeutic” effects in in vitro experiments studying cancer cell proliferation have not yet been successfully implemented for the treatment of patients [24]. 2DG itself reached phase I/II clinical trials for anti-cancer treatment after encouraging animal experiments, but was discontinued for phase 3 clinical trials because no significant effects on tumor growth was observed in patients (https://clinicaltrials.gov/ct2/show/NCT00633087). Furthermore, it has been recently described that targeting hexokinase 2, one of the rate-limiting enzymes in glycolysis, with the natural compound costunolide, was highly efficient in reducing HSC activation in vitro and preventing liver damage in vivo [25,26]. This suggests that inhibiting glycolysis, with better compounds, could be a potential strategy to treat liver fibrosis.

Mitochondrial metabolism is upregulated 5-fold in aHSCs versus qHSCs, so we targeted mitochondrial metabolism by inhibiting glutaminase with CB-839, complex I of the ETC with metformin (in human HSCs) and rotenone (in rat HSCs). CB-839 is a novel glutaminase inhibitor that is currently being evaluated in several clinical trials to test its anti-carcinogenic properties in patients, as it effectively kills cancer cells in in vitro and in vivo models [27,28,29]. Similar as observed for 2DG, CB-839 had different effects in fully activated human and rat HSCs; it significantly suppressed proliferation and expression of fibrogenic markers in human HSCs, while minimal to no effects were observed for rat HSCs. The results using human HSCs are promising, though, and may further support the notion that human HSCs rely more on different energy-generating metabolic processes than rat HSCs. CB-839 was developed to specifically inhibit the glutaminase C (GAC) splice variant of the human glutaminase (GLS1) gene [28], so species differences of this specific splicing variant could also affect the pharmacodynamics of this drug. Interestingly, it has been shown that culturing primary human and rat HSCs in media lacking glutamine in the presence of glucose significantly delays proliferation and activation [8]. The expression of glutaminase is enhanced in chronically injured human livers, providing further support that targeting glutaminolysis with glutaminase inhibitors (such as BPAN and CB-839) may become a successful approach for treating liver fibrosis.

The most important and novel finding of our work was that targeting complex I of the mitochondrial ETC with either metformin (in human HSCs) or rotenone (in rat HSCs) most potently suppressed the proliferation and expression of fibrogenic markers in HSCs. Furthermore, rotenone reduced cellular ATP levels in r-aHSCs, suggesting these cells rely on mitochondrial-derived ATP to proliferate and maintain the activated status. This uncovers mitochondrial OXPHOS as a novel therapeutic target for the treatment of liver fibrosis. Metformin has been in clinical use for the treatment of metabolic disease, particularly type 2 diabetes, for decades already. Interestingly, metformin reduced the mortality rate by 57% in diabetic patients with cirrhotic liver disease [30]. Though this does not provide proof that this is related to an anti-fibrotic effect of metformin, other studies also support this therapeutic action of metformin. For instance, metformin suppressed *COL1A1* expression and induced lipogenic genes in human lung fibroblasts in vitro [31]. Moreover, it reversed already established lung fibrosis in mice [32]. Thus, metformin, and inhibitors of mitochondrial OXPHOS in general, may have therapeutic potential in the treatment of liver fibrosis in chronic liver diseases.

In summary, we show that HSC activation is associated with simultaneous induction of glycolysis and mitochondrial metabolism to support the high energy demand associated with a fibrogenic phenotype. This uncovers novel targets for developing effective therapies to halt and/or reverse liver fibrosis, specifically targeting mitochondrial glutaminolysis or inhibiting the complex I of the ETC.

## Figures and Tables

**Figure 1 cells-09-02456-f001:**
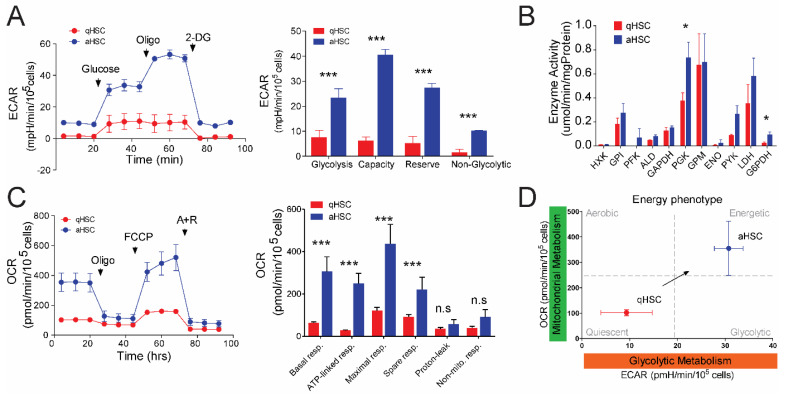
Simultaneous induction of glycolytic and mitochondrial metabolism during hepatic stellate cell (HSC) activation. (**A**) Extracellular acidification rate (ECAR) versus time of quiescent HSCs (qHSCs) (red) and activated HSCs (aHSCs) (blue) after analysis with a real-time glycolysis stress kit to measure glycolytic metabolism; key glycolytic parameters quantified from the data of qHSCs (red) and aHSCs (blue); (**B**) enzymatic activity (Vmax in cell extracts) of key glycolytic enzymes of qHSCs (red) and aHSCs (blue); (**C**) oxygen conspumption rate (OCR) versus time in qHSCs (red) and aHSCs (blue) after real-time analysis with a mitochondrial stress kit to measure mitochondrial metabolism; key mitochondrial metabolism parameters quantified from the data of qHSCs (red) and aHSCs (blue); (**D**) OCR versus ECAR of qHSCs (red) and aHSCs (blue) to create an energy phenotype plot; *n* = 3, mean ± SEM * *p* < 0.05, *** *p* < 0.0001 (one-way ANOVA).

**Figure 2 cells-09-02456-f002:**
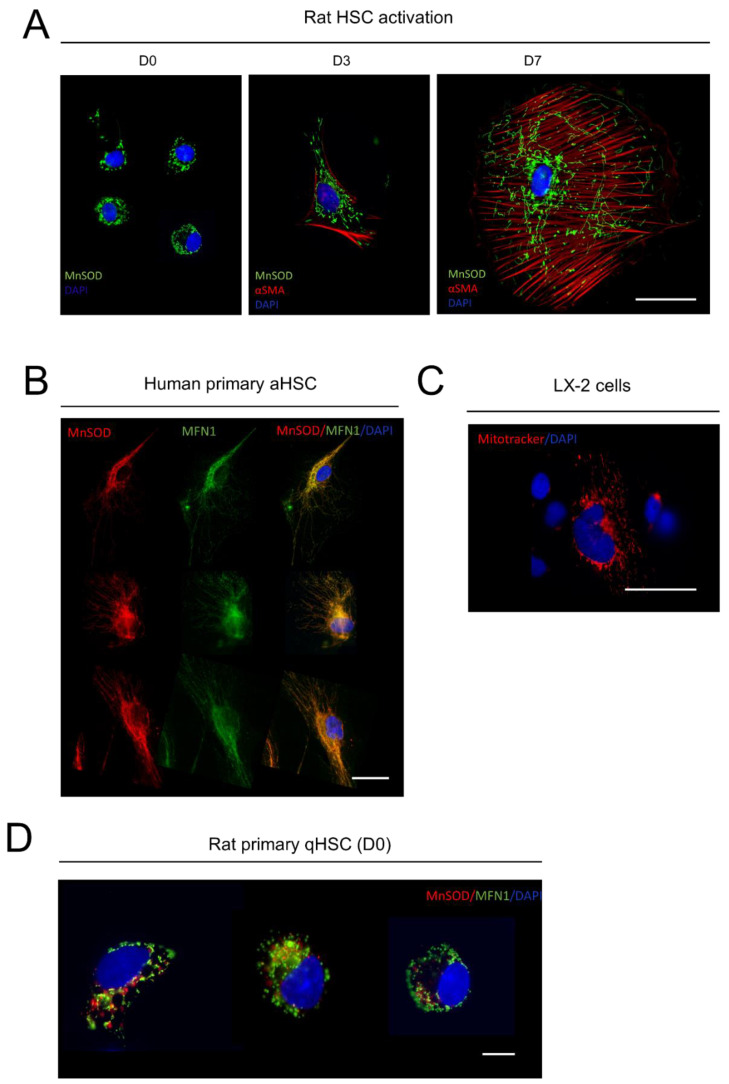
Mitochondrial fusion is increased during activation of HSCs. (**A**) Immunofluorescence of HSC activation (4 h, 3 d and 7 days after isolation) to assess the mitochondrial enzyme manganese superosxidase (MnSOD, in green), and the activation marker, alpha-smooth muscle actin (α-SMA) (in red). Nuclei were stained with DAPI (in blue). Representative pictures from three independent experiments. Bar = 50 µm. (**B**) Immunofluorescence of human aHSCs to assess MnSOD (in red), and the mitochondrial fusion protein mitofusin 1 (MFN1, in green), and DAPI (in blue). Bar = 50 µm. (**C**) Immunofluorescence staining of LX-2 cells using MitoTracker (in red) and DAPI (in blue). Bar = 50 µm. (**D**) Immunofluorescence staining of rat qHSCs to assess mitochondria using MnSOD (in green) and MFN1 (in red), nuclei were stained with DAPI (in blue). Bar = 10 µm.

**Figure 3 cells-09-02456-f003:**
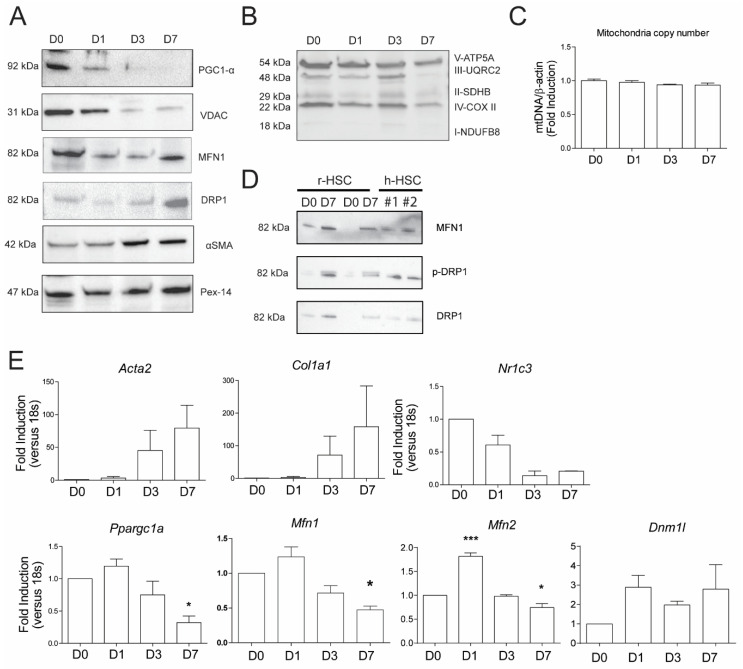
Increase in mitochondrial fusion during activation is independent of mitochondrial biogenesis or mitochondrial copy number: (**A**) Western blots of day 0, day 1, day 3 and day 7 after isolation of HSCs against PGC1-α (mitochondrial biogenesis), voltage-dependent anionic channel (VDAC) (marker of mitochondrial mass), MFN1 and dynamin-related protein 1 (DRP1) (markers of mitochondrial fusion), α-SMA (HSC activation) and Pex-14 (loading control); (**B**) Western blots of day 0, day 1, day 3 and day 7 after isolation of HSCs to measure the electron transport chain (ETC) complexes; (**C**) mitochondrial copy number quantified by relative DNA expression of mitochondrial gene NDH2 to genomic DNA (β-actin) upon activation of HSCs; (**D**) Western blots of rat HSCs (day 0 and day 7) and human aHSCs against MFN1, DRP1, p-DRP1; (**E**) relative mRNA expression of α-SMA (encoded by *acta2*), collagen I (encoded by *col1a1*), PPARγ (encoded by *pparγ*), PGC1α (encoded by *ppargc1a*), MFN1 and MFN1 (encoded by *Mfn1* and *MFN2*, respectively) and DRP1 (encoded by *Dnm1l*) versus *18S*; *n* = 3, mean ± SEM * *p* < 0.05, *** *p* < 0.0001 (one-way ANOVA).

**Figure 4 cells-09-02456-f004:**
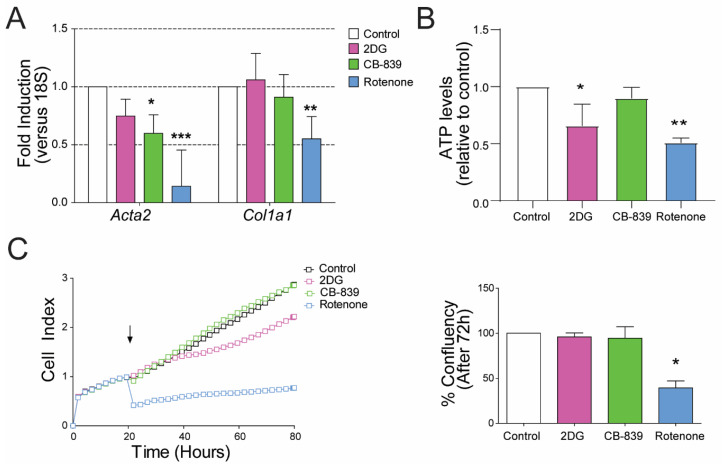
Glutaminolysis and oxidative phosphorylation (OXPHOS) are independently necessary for maintaining HSC activated phenotype. Primary rat activated HSCs were treated with glycolysis inhibitor 2-Deoxy-D-glucose (2DG), glutaminase inhibitor CB-839 and complex I of electron transport chain (ETC) inhibitor, rotenone for 72h. (**A**) The mRNA of α-SMA (*Acta2*) and collagen I (*Col1a1*), (**B**) cellular ATP concentrations relative to control and (**C**) real-time proliferation (arrow indicates the start of the treatment) and bar graph summarizing the confluency percentage after 3 days of treatment calculated using an automated cell imager and corresponding software (Incucyte, Zoom); *n* = 3, mean ± SEM * *p* < 0. 05, ** *p* < 0.001, *** *p* < 0.0001 (one-way ANOVA).

**Figure 5 cells-09-02456-f005:**
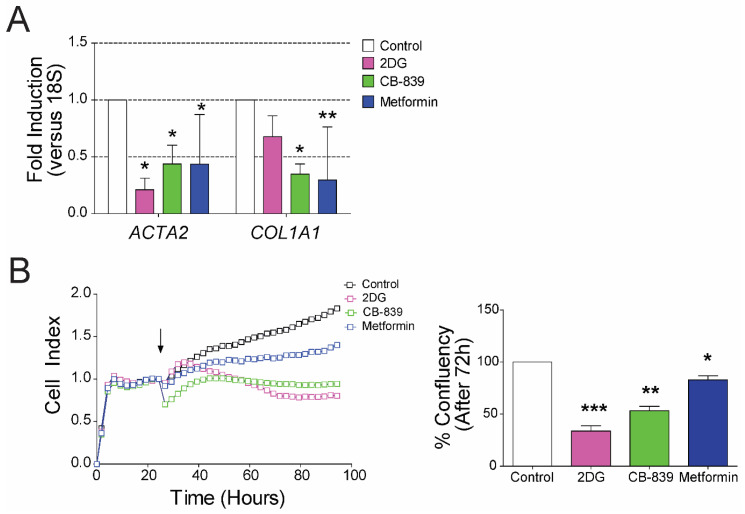
Glycolysis, glutaminolysis and OXPHOS are independently necessary for maintaining HSC activated phenotype. Primary human activated HSCs were treated with glycolysis inhibitor 2DG (1 mmol/L), glutaminase inhibitor CB-839 (5 µmol/L) and complex I of ETC inhibitor, metformin (2 mmol/L) for 72 h. (**A**) The mRNA of aSMA (*ACTA2*) and collagen I (*COL1A1*), (**B**) real-time proliferation (arrow indicates the start of the treatment) and bar graph summarizing the confluency percentage after 3 days of treatment calculated using an automated cell imager and corresponding software (Incucyte, Zoom); *n* = 3, mean ± SEM * *p* < 0.05, ** *p* < 0.001, *** *p* < 0.0001 (one-way ANOVA).

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
