# Peer review of "Simultaneous Induction of Glycolysis and Oxidative Phosphorylation during Activation of Hepatic Stellate Cells Reveals Novel Mitochondrial Targets to Treat Liver Fibrosis"

_cells, 2020, doi:10.3390/cells9112456_

Round 1

Reviewer 1 Report

Aim of this work was to explore the contribution of glycolysis and OXPHOX in rat and human hepatic stellate cells (HSC) activation. Authors conclude that simultaneous induction of both processes is required for HSC activation, and propose that interfering with HSC metabolism could be a therapeutic option in hepatic fibrogenesis. Novelty of the study is under question and results are very descriptive and preliminary in some aspects. Further work is necessary before publication of the manuscript.

MAJOR POINTS

  1. Authors conclude that mitochondrial fusion is increased during activation of HSC, but only a immunocytochemistry image is shown. Quantification of the mitochondrial lenght, circularity, branches, etc is necessary and authors could obtain these results with the adequate software. Statistical analysis must be performed. Furthermore, authors must analyse the expression of proteins related to mitochondria fusion (mitofusin family, among others).
  2. Why do authors use rat and human cells alternatively in the different experiments? Whis is the reason to use rotenone only in rat and metformin in human? 
  3. Figure 4: Effects of inhibiting glutaminolysis in rat activated HSC is not drastic: no effects on Col1a1 expression and barely decrease in Acta2 and no effect on cell index. These experiments do not demonstrate the relevance of glycolyisis or glutaminolysis. 
  4. Results in human cells and with metformin are more convincing. However, the OCR and ECAR analysis shown in Fig. 1 was performed in rat HSC. A similar analysis must be performed here.
  5. In both Fig. 4 and 5, the effect of 2DG , CB839 and rotenone or metformin on OCR and ECAR is necessary to demonstrate the efficiency of the inhibitors on glucolysis and OXPHOX. 
  6. A relevant analysis has been lost in all the study, which is the analysis of ATP levels along the HSC activation process and by the treatment with the respective inhibitors.

MINOR POINTS:

  1. Introduction must better explain which is the novelty of the study. The previous report indicating that glutaminolysis is required for the high energy demand for HSC activation demonstrates the requirement of TCA (and potentially OXPHOX), where glutamine is incorporated. Glycolisis requirement was also previously reported, as authors mention.
  2. How do authors explain the loss of PGCi-alpha and VDAC along time during HSC activation, whereas mitochondria copy number and OHPHOX proteins do not change? 
  3. References lack information in some cases (#8, #22, #27). In others, include [internet] in the middle of the reference that is not necessary. Please, revise all the citations.

Author Response

Reviewer 1:

Aim of this work was to explore the contribution of glycolysis and OXPHOX in rat and human hepatic stellate cells (HSC) activation. Authors conclude that simultaneous induction of both processes is required for HSC activation, and propose that interfering with HSC metabolism could be a therapeutic option in hepatic fibrogenesis. Novelty of the study is under question and results are very descriptive and preliminary in some aspects. Further work is necessary before publication of the manuscript.

MAJOR POINTS

  1. Authors conclude that mitochondrial fusion is increased during activation of HSC, but only a immunocytochemistry image is shown. Quantification of the mitochondrial lenght, circularity, branches, etc is necessary and authors could obtain these results with the adequate software. Statistical analysis must be performed. Furthermore, authors must analyse the expression of proteins related to mitochondria fusion (mitofusin family, among others).

Author reply: We thank the reviewer for this comment that motivated us to include additional data on mitochondrial biogenesis. In the revised Figure 2, we include additional images of day 0 r-qHSC (Figure 2A and D), where we can clearly observe dotted-like mitochondria as compared to day 7 r-HSC where mitochondria are highly fused. In addition, we include MnSOD/aSMA costaining at day 3 (Figure 2A) of activation that shows the initial stages formation of a mitochondrial network. Moreover, we include immunofluorescence staining of MnSOD and MFN1 in primary human aHSC (Figure 2B) and quiescent rat HSC (Figure 2D) and mitotracker staining of LX-2 cells (Figure 2C). Unfortunately, the limited magnification and resolution of our confocal microscope images do not allow an adequate quantification of mitochondrial morphology in the short time frame given for resubmitting our manuscript. We hope that our revised Figure 2 will convince the reviewer of the clear mitochondrial changes associated with HSC activation.

2.1 Why do authors use rat and human cells alternatively in the different experiments?

Author reply: We used rat HSC to compare freshly-isolated quiescent q-HSC versus culture-activated aHSC, as the difference in mitochondrial morphology and activity (OXPOS) in freshly-isolated qHSC vs aHSC has not been demonstrated before.  We were not able to compare human qHSC vs aHSC, as the isolation and purification protocol of HSC from healthy parts of resected (tumor-containing) liver tissue includes a culturing phase where plastic-adherent HSC are enriched over other contaminating hepatic cell types (such as hepatocytes and endothelial cells). The human HSC rapidly activate in this culturing step. As a consequence, we could only analyze activated human HSC in our study.

2.2 Whis is the reason to use rotenone only in rat and metformin in human? 

Author reply: Rotenone is the typical compound to effectively block mitochondrial OXPHOS so we used it as a proof of concept in rat HSC. However, rotenone is also known to be very toxic. Thus, for making a translation to clinically relevant therapies, we used metformin to inhibit mitochondrial OXPHOS in human HSC.

3. Figure 4: Effects of inhibiting glutaminolysis in rat activated HSC is not drastic: no effects on Col1a1 expression and barely decrease in Acta2 and no effect on cell index. These experiments do not demonstrate the relevance of glycolysis or glutaminolysis. 

Author reply: We agree with the reviewer that there is little/no significant effect of inhibiting glutaminolysis in rat HSC, which is clearly different in human HSC. More emphasis is put on this difference in the revised discussion.

4. Results in human cells and with metformin are more convincing. However, the OCR and ECAR analysis shown in Fig. 1 was performed in rat HSC. A similar analysis must be performed here.

Author reply: We fully understand this suggestion of the reviewer. However, as our isolation protocol is not suitable to obtain quiescent human HSC (see also rebuttal of point 2) we could only perform this analysis with rat HSC. A previous study analyzed OCR and ECAR in human LX-2 cells (a commonly used HSC cell line that we also included in our study) when they were grown on plastic (resembling activated HSC) or on matrigel (leading to suppression of the fibrogenic phenotype) (Gajendiran et al., 2018, see also reviewer 2). In line with our data with rat HSC, they show that OCR and ECAR are both strongly increased in on plastic-cultured LX-2 cells compared to on matrigel-grown cells. We include a description of these results in our revised manuscript in support of our own findings.

5. In both Fig. 4 and 5, the effect of 2DG , CB839 and rotenone or metformin on OCR and ECAR is necessary to demonstrate the efficiency of the inhibitors on glycolysis and OXPHOX. 

Author reply: Given the fact that we see effects of these compounds on the fibrogenic phenotype of HSC (proliferation and activation markers) using concentrations also used by others to inhibit OCR and ECAR,we feel it is not essential for our message to confirm this independently. Moreover, the effects of the deprivation of either glucose or glutamine in rat aHSC OCR has already been described by Du K, Hyun J, Premont RT, Choi SS, Michelotti GA, Swiderska-Syn M, et al. Hedgehog-YAP Signaling Pathway Regulates Glutaminolysis to Control Activation of Hepatic Stellate Cells. Gastroenterology Apr;154(5):1465-1479.e13..

6. A relevant analysis has been lost in all the study, which is the analysis of ATP levels along the HSC activation process and by the treatment with the respective inhibitors.

Author reply: We do agree that analyzing the ATP levels would be relevant, but is not essential for the current message of our work. Moreover, the limited time given for submitting our revised manuscript (10 days) did not allow us to perform this analysis.

MINOR POINTS:

1.Introduction must better explain which is the novelty of the study. The previous report indicating that glutaminolysis is required for the high energy demand for HSC activation demonstrates the requirement of TCA (and potentially OXPHOX), where glutamine is incorporated. Glycolisis requirement was also previously reported, as authors mention.

Author reply: We have revised the introduction in order to better explain the novelty of our findings.

2.How do authors explain the loss of PGCi-alpha and VDAC along time during HSC activation, whereas mitochondria copy number and OHPHOX proteins do not change? 

Author reply: We agree with the reviewer that this is a puzzling observation. PGC1-alpha is a co-activator of the PPARgamma, which is a transcription factor that is markedly downregulated in the activation of HSC. The role of PGC1-alpha in mitochondrial biogenesis, however, is mostly driven by its role as co-activator of nuclear respiratory factor 2 (NRF2). Given the fact that PPARgamma is virtually absent in aHSC, the reduced levels of PGC1-alpha may therefore remain sufficient to activate NRF2 and maintain mitochondrial copy number. The effect in VDAC is more difficult to explain. We expected to see no changes in this marker since we did not see an increase in mitochondrial mass, but maybe turnover of VDAC is regulated independently from total mitochondrial turnover. We include these possibilities in the revised discussion.

3. References lack information in some cases (#8, #22, #27). In others, include [internet] in the middle of the reference that is not necessary. Please, revise all the citations.

Author reply: We thank the reviewer for pointing out these textual errors, for which we apologize. We have corrected the lack of information (#8, line 483, #22, line 512, #27, line 523) and removed the word [internet] in the middle of some references.

Reviewer 2 Report

The authors in this manuscript submission present a study that compared the contribution of glycolysis, glutaminolysis and mitochondrial oxidative phosphorylation (OXPHOS) in rat and human hepatic stellate cells (HSC) activation. Authors show that the basal levels of glycolysis (extracellular acidification rate, ECAR) and mitochondrial respiration (oxygen consumption rate, OCR) were significantly increased in rat activated HSC (aHSC), when compared to quiescent rat HSC (qHSC). This was accompanied by extensive mitochondrial fusion in r-aHSC, which occurred without increasing mitochondrial DNA content and electron transport chain components. Inhibition of glycolysis (by 2-deoxy-D-glucose) and glutaminolysis (by CB-839) did not inhibit r-aHSC proliferation, but did reduce Acta2 (encoding αSMA) expression slightly. In contrast, inhibiting mitochondrial OXPHOS significantly suppressed r-aHSC proliferation, as well as Col1a1 and Acta2 expression. In human activated HSC (h-aHSC) proliferation and expression of fibrosis markers were significantly suppressed by inhibiting either glycolysis, glutaminolysis or mitochondrial OXPHOS. Authors conclude that the activation of HSC is marked by simultaneous induction of glycolysis and mitochondrial metabolism, extending the possibilities to suppress hepatic fibrogenesis by interfering with HSC metabolism.

The current study does seem to expand upon the previous studies that showed elevated mitochondrial activity that distinguishes activated fibrogenic HSC from quiescent HSC in both rat and human (Gajendiran et al, J Cell Mol Med Vol 22, No 4, 2018 pp. 2210-2219, Ref#7, 26-28). However, it is surprising that Gajendiran study was not cited in the manuscript. In Gajendiran et al. study, the analysis of mitochondrial respiration showed a striking elevation in the OCR of fibrogenic HSCs compared to the normal, less-active counterpart as demonstrated from human HSC, LX-2 and rat HSCs. The significant increase in mito-respiration also reflected in the ECAR. Energy map based on the OCR and ECAR demonstrated that fibrogenic HSCs have higher energy level (bioenergetic capacity) perhaps to corroborate the functional demand. The fibrogenic HSCs have markedly increased mito-membrane potential compared to the less active HSCs of both human and rat origin. While there are some merits (use of new metabolic targets to treat liver fibrosis) in this current study under review, the novelty of this study is not clear in light of Gejendiran et al and other studies. Perhaps, this current study needs to emphasize more on the metabolic targets i.e. 2DG, CB-839, rotenone and metformin as the possible targets to treat liver fibrosis.

Major Concerns:

1. Authors show glycolytic and mitochondrial metabolism markers such as ECAR, OCR (Fig. 1) and mitochondrial fusion MnSod (Fig. 2) in qHSC vs aHSC from rat. It will be a good idea to measure/show these markers as the activation of HSC progressing in longitudinal time points such as day 1, day 3, and day 5. This may help to understand how these markers differentiate during the activation of HSC in real time. Moreover, did authors measure/show these markers in human h-qHSC vs h-aHSC during the activation of HSC?

2. Interestingly, authors show that the inhibition of glycolysis caused a significant reduction in ACTA2 (encoding α-SMA) and COL1A1 (encoding collagen-I) mRNA levels only in human aHSC (Figure 5A) with no effect in r-aHSC (Figure 4A). In line, similar effects were observed h-aHSC proliferation. Targeting glutaminolysis had also differential effects in r-aHSC and h-aHSC; While ACTA2/Acta2 levels were suppressed by CB-839 treatment in both r-aHSC and h-aHSC, COL1A1 levels were only reduced in h-aHSC (Figures 4A and 5A). In light of Chen et al study (Ref#7), it seems that inhibiting glycolysis at early time points might be beneficial to inhibit HSC activation in rat, whereas in humans it can be inhibited even after full activation. This is really an intriguing observation, and may need further attention. This could be species-dependent as authors suggest. Do concentrations of 2-DG and durations of treatment make difference? How does this compare to Costunolide which has been shown as the potent inhibitor of HSC activation thorough inhibition of aerobic glycolysis (Ban et al, Cellular and Molecular Biology Letters 2019; 24:52). Perhaps, measuring ECAR and other markers in h-qHSC vs h-aHSC during the activation of HSC with treatment in real time may shed light onto this matter.

3. Similar as observed for 2DG, CB-839 had different effects in fully activated human and rat HSC; it significantly suppressed proliferation and expression of fibrogenic markers in human HSC, while minimal-to-no effects were observed for rat HSC. Authors conclude that the results using human HSC are promising, though, and may further support the notion that human HSC rely more on different energy-generating metabolic processes than rat HSC. This needs more attention as suggested in point 2.

4. The important observation as authors highlight is that targeting complex I of the mitochondrial ETC with either metformin (in human HSC) or rotenone (in rat HSC) most potently suppressed the proliferation and expression of fibrogenic markers in HSC. This uncovers mitochondrial OXPHOS as a novel therapeutic target for the treatment of liver fibrosis. OXPHOS as therapeutic target has been studied by many research groups (Ref# 26-28), and the inhibition of OXPHOS were effective in both human and rat HSC in this study as well. Perhaps authors should emphasize and highlight this pathway in the manuscript.

Author Response

Reviewer 2:

The authors in this manuscript submission present a study that compared the contribution of glycolysis, glutaminolysis and mitochondrial oxidative phosphorylation (OXPHOS) in rat and human hepatic stellate cells (HSC) activation. Authors show that the basal levels of glycolysis (extracellular acidification rate, ECAR) and mitochondrial respiration (oxygen consumption rate, OCR) were significantly increased in rat activated HSC (aHSC), when compared to quiescent rat HSC (qHSC). This was accompanied by extensive mitochondrial fusion in r-aHSC, which occurred without increasing mitochondrial DNA content and electron transport chain components. Inhibition of glycolysis (by 2-deoxy-D-glucose) and glutaminolysis (by CB-839) did not inhibit r-aHSC proliferation, but did reduce Acta2 (encoding αSMA) expression slightly. In contrast, inhibiting mitochondrial OXPHOS significantly suppressed r-aHSC proliferation, as well as Col1a1 and Acta2 expression. In human activated HSC (h-aHSC) proliferation and expression of fibrosis markers were significantly suppressed by inhibiting either glycolysis, glutaminolysis or mitochondrial OXPHOS. Authors conclude that the activation of HSC is marked by simultaneous induction of glycolysis and mitochondrial metabolism, extending the possibilities to suppress hepatic fibrogenesis by interfering with HSC metabolism.

The current study does seem to expand upon the previous studies that showed elevated mitochondrial activity that distinguishes activated fibrogenic HSC from quiescent HSC in both rat and human (Gajendiran et al, J Cell Mol Med Vol 22, No 4, 2018 pp. 2210-2219, Ref#7, 26-28). However, it is surprising that Gajendiran study was not cited in the manuscript. In Gajendiran et al. study, the analysis of mitochondrial respiration showed a striking elevation in the OCR of fibrogenic HSCs compared to the normal, less-active counterpart as demonstrated from human HSC, LX-2 and rat HSCs. The significant increase in mito-respiration also reflected in the ECAR. Energy map based on the OCR and ECAR demonstrated that fibrogenic HSCs have higher energy level (bioenergetic capacity) perhaps to corroborate the functional demand. The fibrogenic HSCs have markedly increased mito-membrane potential compared to the less active HSCs of both human and rat origin. While there are some merits (use of new metabolic targets to treat liver fibrosis) in this current study under review, the novelty of this study is not clear in light of Gejendiran et al and other studies. Perhaps, this current study needs to emphasize more on the metabolic targets i.e. 2DG, CB-839, rotenone and metformin as the possible targets to treat liver fibrosis.

Author reply: We deeply apologize for not properly mentioning and citing the paper of Gajendiran et al, which indeed is of great relevance for our manuscript. Accordingly, we now cite this paper in relevant sections of the introduction and discussion of the revised manuscript. In light of that study, we do feel however that our work still has important novelty, as they analyzed OCR and ECAR in culture-activated rat HSC and human LX-2 cells in comparison to the same cells grown on matrigel-coated surfaces. Although this is an accepted method to suppress the fibrogenic phenotype of HSC, it may not fully resemble true quiescent HSC in a healthy liver (which are, for instance, highly active in controlling vitamin A metabolism). Thus, we feel that the analysis of the freshly-isolated rat HSC clearly adds novelty to the current knowledge, even though we observe similar effects on OCR and ECAR.

In addition, we put more emphasis on the metabolic targets in the revised manuscript, as suggested by this reviewer.

Major Concerns:

1. Authors show glycolytic and mitochondrial metabolism markers such as ECAR, OCR (Fig. 1) and mitochondrial fusion MnSod (Fig. 2) in qHSC vs aHSC from rat. It will be a good idea to measure/show these markers as the activation of HSC progressing in longitudinal time points such as day 1, day 3, and day 5. This may help to understand how these markers differentiate during the activation of HSC in real time. Moreover, did authors measure/show these markers in human h-qHSC vs h-aHSC during the activation of HSC?

Author reply: Following the suggestion of the reviewer, we included immunofluorescence staining of MnSOD at day 3 of activation of rat HSC (revised Figure 2A, and added to the D0 and D7 rat HSC) to show intermediate morphological changes  in mitochondria in transactivating rat HSC. In addition, we included immunofluorescence staining of MnSOD and MFN1 in human aHSC (Figure 2B) and mitotracker staining of LX-2 cells (Figure 2C) to demonstrate that fused mitochondrial networks are also observed in activated human HSC. Again (see also rebuttal reviewer 1), it would have been great to include similar data for human HSC, however, the isolation and purification protocol of human HSC used in this study only allows for the analysis of fully activated HSC, so a comparison to quiescent HSC could not be made. For this, we now also refer to Gajendiran et al., who showed that de-activation of human HSC leads to a similar decrease in OCR and ECAR.

2. Interestingly, authors show that the inhibition of glycolysis caused a significant reduction in ACTA2 (encoding α-SMA) and COL1A1 (encoding collagen-I) mRNA levels only in human aHSC (Figure 5A) with no effect in r-aHSC (Figure 4A). In line, similar effects were observed h-aHSC proliferation. Targeting glutaminolysis had also differential effects in r-aHSC and h-aHSC; While ACTA2/Acta2 levels were suppressed by CB-839 treatment in both r-aHSC and h-aHSC, COL1A1 levels were only reduced in h-aHSC (Figures 4A and 5A). In light of Chen et al study (Ref#7), it seems that inhibiting glycolysis at early time points might be beneficial to inhibit HSC activation in rat, whereas in humans it can be inhibited even after full activation. This is really an intriguing observation, and may need further attention. This could be species-dependent as authors suggest. Do concentrations of 2-DG and durations of treatment make difference? How does this compare to Costunolide which has been shown as the potent inhibitor of HSC activation thorough inhibition of aerobic glycolysis (Ban et al, Cellular and Molecular Biology Letters 2019; 24:52). Perhaps, measuring ECAR and other markers in h-qHSC vs h-aHSC during the activation of HSC with treatment in real time may shed light onto this matter.

Author reply: We fully agree with the reviewer that it is intriguing that fully-activated human HSC seem more responsive to inhibition of glycolysis, glutaminolysis and OXPHOS when compared to fully-activated rat HSC. Indeed, Chen et al (ref #7) started 2DG treatments with HSC in culture for only 3 days which are not full-activated yet, and thus may indicate that 2DG may have most prominent effects on rat HSC that are in the process of transdifferentiation. Our data indicate that fully-activated rat HSC are less sensitive to 2DG treatment, while human HSC are. We thank the reviewer for pointing out the paper by Ban et al (Cellular and Molecular Biology Letters 2019; 24:52). Indeed, this compound as anti-fibrotic effects in vitro at much lower concentrations than 2DG (10-30 uM versus 2.5-10 mM, respectively) and has been shown to have hepatoproctive effects in rodent models acute and chronic liver disease. However, inhibition of aerobic glycolysis may not be the (only/primary) therapeutic action, as it also acts as a potent anti-oxidant. We make a short state on this in revised discussion. As a final note (see also rebuttal point 1/reviewer 1), our study did not allow a comparison between quiescent and culture-activated human HSC, unfortunately.

3. Similar as observed for 2DG, CB-839 had different effects in fully activated human and rat HSC; it significantly suppressed proliferation and expression of fibrogenic markers in human HSC, while minimal-to-no effects were observed for rat HSC. Authors conclude that the results using human HSC are promising, though, and may further support the notion that human HSC rely more on different energy-generating metabolic processes than rat HSC. This needs more attention as suggested in point 2.

Author reply: We thank the reviewer for this suggestion. We have to be careful to directly compare the effects of CB-839 in rat and human HSC, as this compound was synthetized to target specifically the human glutaminase isoform GLS1. Thus, we cannot rule out species specificity here. We do mention this in the revised manuscript, but follow the reviewer in hypothesizing that activated human HSC indeed rely more on energy-generating metabolic processes than rat HSC

4. The important observation as authors highlight is that targeting complex I of the mitochondrial ETC with either metformin (in human HSC) or rotenone (in rat HSC) most potently suppressed the proliferation and expression of fibrogenic markers in HSC. This uncovers mitochondrial OXPHOS as a novel therapeutic target for the treatment of liver fibrosis. OXPHOS as therapeutic target has been studied by many research groups (Ref# 26-28), and the inhibition of OXPHOS were effective in both human and rat HSC in this study as well. Perhaps authors should emphasize and highlight this pathway in the manuscript.

Author reply: We thank the reviewer for this suggestion and we have further emphasized that our work especially identifies HSC OXPHOS as a potential target to treat liver fibrosis (for instance using metformin).

Round 2

Reviewer 1 Report

I congratule authors for the additional experiments performed. However, I still consider that ATP levels must be analyzed. Otherwise, it is not possible to understand the impact of the metabolic changes observed. I also recommed to clarify in the manuscript the reasons for the use of both rat and human stellate cells, such as explained in the answers to my concerns. 

Author Response

Author’s reply: We thank the reviewer for his/her very positive assessment of the first revision of our manuscript. We agree that analyzing ATP levels indeed has additional value for the interpretation of our experiments. Thus, as requested by the reviewer, we have quantified ATP levels in the experiments where we exposed rat HSC to the various inhibitors of cellular/mitochondrial metabolism, e.g. 2DG, CB-839 or Rotenone. As you can see, ATP levels are reduced after 72 h in primary rat HSC treated with 2DG and Rotenone. CB-839 did not cause a strong reduction in ATP levels, probably because cells have efficient compensatory mechanisms, such as increasing glycolysis. Interestingly, ATP levels nicely correlated with the effect of the inhibitors on HSC proliferation. We have added this experiment in Figure 4 (new panel 4B of the revised manuscript). Currently, our research activities are limited by covid-19 related restrictions, so we have not been able to also perform such experiments for human HSC. We feel that our experiments with rat HSC show that the concentrations of the inhibitors used indeed effect ATP levels and correlate well with the effect on cell proliferation. We hope that these results (which are included in the revised manuscript) will convince you to make our manuscript acceptable for publication.

Finally, we have clarified the use of rat and human HSC for specific experiments in the discussion section, using the same phrasing in our answers to your earlier comments.